# REWARD COLLAPSE IN ALIGNING LARGE LANGUAGE MODELS

## ABSTRACT

The extraordinary capabilities of large language models (LLMs) such as ChatGPT and GPT-4 are in part unleashed by aligning them with reward models that are trained on human preferences represented as rankings of responses to prompts. In this paper, we document the phenomenon of *reward collapse*, an empirical observation where the prevailing ranking-based approach results in an *identical* reward distribution for diverse prompts during the terminal phase of training. This outcome is undesirable as open-ended prompts like "write a short story about your best friend" should yield a continuous range of rewards for their completions, while specific prompts like "what is the capital city of New Zealand" should generate either high or low rewards. Our theoretical investigation reveals that reward collapse is primarily due to the insufficiency of the ranking-based objective function to incorporate prompt-related information during optimization. This insight allows us to derive closed-form expressions for the reward distribution associated with a set of utility functions in an asymptotic setting. To overcome reward collapse, we introduce a prompt-aware optimization scheme that provably admits a prompt-dependent reward distribution within the interpolating regime. Our experimental results suggest that our proposed prompt-aware utility functions significantly alleviate reward collapse during the training of reward models.

## 1 INTRODUCTION

A cornerstone of the recent remarkable advancements in the capabilities of large language models (LLMs) like ChatGPT and GPT-4 is the integration of human feedback (Ouyang et al. (2022); OpenAI (2023)). The approach to leveraging human feedback often begins with the training of a reward model that encapsulates human preferences, values, and ethical considerations (Christiano et al. (2017); Ibarz et al. (2018); Bahdanau et al. (2018); Ziegler et al. (2019); Ganguli et al. (2022)). This is followed by the fine-tuning of the LLMs using reinforcement learning, guided by the reward model. This process, often referred to as reinforcement learning from human feedback (RLHF), has proven effective in aligning LLMs with human intent, substantially enriching the quality of human interaction.

However, developing an effective reward model based on human preferences is challenging (Bai et al. (2022b); Liu et al. (2023); Sun et al. (2023)). A notable difficulty arises when a human labeler struggles to give a quantitative score to a response/completion for a specific prompt. Instead, it is much easier for humans to make pairwise comparisons between completions in terms of their quality, which is indeed employed in the development of InstructGPT (Ouyang et al. (2022)). Explicitly, a human labeler is presented with several completions generated by the LLMs for the same prompt and arranges the responses from the highest to lowest perceived quality.[1] A neural network is then trained to obtain a reward model that assigns rewards to the responses in an attempt to align as closely as possible with human preferences in the form of *rankings*.

Despite some benefits, such as eliminating calibration issues, rankings fall short in reflecting the varied reward distributions of different prompts. This is due to the fact that ranking one completion higher than another does not indicate how *much* superior the former is compared to the latter. This

---

[1]In slightly more detail, Ouyang et al. (2022) required human labelers to utilize a drag-and-drop interface to construct *consistent* rankings from pairwise comparisons.

concern is especially pertinent in RLHF as some prompts are open-ended or, in other words, are dependent on the users' backgrounds, allowing the reward distribution to span a continuous range. Conversely, some prompts are closed-ended, resulting in a response that should be either highly or lowly scored, thus generating a roughly two-point mass distribution for the reward distribution. Instances of the first type of prompts include *write a short story about how AI will look like in 100 years* and *what is the best cuisine in the world*, while examples of the second type are *prove the Pythagorean theorem* and *is chicken a dinosaur*. An ideal reward model would assign a reward of either low or high to close-ended prompts, ensuring that the completion accurately aligns with the correct direction. Conversely, for open-ended prompts, the reward should avoid being either low or high to encourage diverse responses. If the reward model cannot distinguish between open-ended and close-ended prompts, it fails to assist language models in determining uncertainty when providing completions, whether with high variability or low variability (Padmakumar & He (2023)). As a result, the reward model may struggle to aid LLMs in accurately calibrating uncertainty without accounting for the nuances of different prompts. [2]

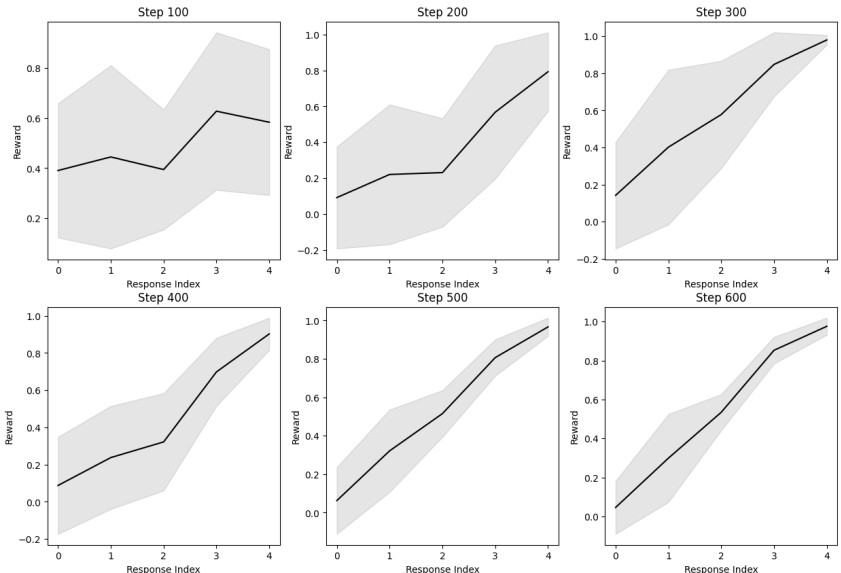

Figure 1: Evidence of reward collapse in an Large language model. Experiment details are elaborated in Section 3.

As our first main contribution, this paper documents a surprising phenomenon through a series of experiments, demonstrating that training a reward model on preference rankings could result in the *same* reward distribution regardless of the prompts. We call this phenomenon *reward collapse*, which occurs during the terminal phase of training Papyan et al. (2020). Intriguingly, our theoretical analysis first predicted this phenomenon prior to its experimental confirmation. Indeed, we show that the collapse reward distribution can be numerically deduced from a simple optimization program or, even simpler, admits a closed-form expression. As demonstrated in Figure 1, our prediction of reward collapse is in agreement with the empirical results.

Reward collapse is clearly undesirable as it overlooks the subtle differences among various prompts, potentially leading to the miscalibration of human preference during the training of LLMs via reinforcement learning with the reward model. A rudimentary strategy to bypass this issue is to early stop the training of the reward model (Ouyang et al. (2022)), which, however, is somewhat arbitrary and can make it challenging to determine the stopping point.

In our second main contribution, we introduce a principled approach to alleviating reward collapse, leveraging insights derived from the same optimization program that was instrumental in predicting this phenomenon. In essence, we propose to use distinct utility functions depending on prompts in

---

[2]For instance, we suspect that this is partly accountable for the poor calibration of GPT-4 after RLHF (see page 12 of OpenAI (2023)) and mode collapse (Casper et al. (2023a;b)).

training the reward model, such that the resulting reward distribution can be either widely dispersed or tightly concentrated, contingent on whether the prompt is open-ended or closed-ended. A notable advantage of this prompt-aware strategy is that our analysis is analytical, enabling full control over the shape of the reward distribution as required. Our experiments show that reward collapse can be substantially mitigated using this prompt-aware methodology.

## 2 WHAT IS REWARD COLLAPSE AND HOW TO MITIGATE IT?

### 2.1 REWARD COLLAPSE

We use $\texttt{prom}$ and $\texttt{compl}$ to denote a prompt and a completion. Denote by $R(\texttt{prom}, \texttt{compl})$ a reward model. Without loss of generality, we assume $R(\texttt{prom}, \texttt{compl}) \in [0, 1]$. For a given prompt and $n$ completions that are i.i.d. draws from an LLM, a human labeler ranks the $n$ responses from the most preferred to the least preferred, and the ranking is denoted as $\pi_{\texttt{prom}}$. The reward model is expected to score each completion that is consistent with the human-provided ranking $\pi_{\texttt{prom}}$ as much as possible. To this end, we train a neural network that maximizes the following overall utility:

$$\sum_{(\texttt{prom}, \texttt{compl}_w, \texttt{compl}_l) \in \Pi} U\left(R_\theta(\texttt{prom}, \texttt{compl}_w) - R_\theta(\texttt{prom}, \texttt{compl}_l)\right), \qquad (1)$$

where $U$ is an (increasing) utility function, $\theta$ is the weights of the reward neural network, and $\Pi$ is the ranking dataset and $\texttt{compl}_w$ is a preferred completion than $\texttt{compl}_l$ in the ranking $\pi_{\texttt{prom}}$. In InstructGPT (Ouyang et al. (2022)), $U$ is set to $U_\sigma(x) = \log \texttt{sigmoid}(x/\sigma) \equiv \log \frac{e^{x/\sigma}}{e^{x/\sigma}+1}$, which is an increasing concave function. While maximizing Eq. 1, the reward model learns to not only align with the human-provided ranking but also distinguish the rewards as much as possible.

To gain insights into how the rewards depend on $U$, note that the above is equivalent to

$$\max \sum_{\texttt{prom}} \sum_{(\texttt{compl}_w, \texttt{compl}_l) \in \pi_{\texttt{prom}}} U\left(R_\theta(\texttt{prom}, \texttt{compl}_w) - R_\theta(\texttt{prom}, \texttt{compl}_l)\right).$$

Next, assume that the neural network parameterized by $\theta$ is sufficiently overparameterized such that

$$\sum_{(\texttt{compl}_w, \texttt{compl}_l) \in \pi_{\texttt{prom}}} U\left(R_\theta(\texttt{prom}, \texttt{compl}_w) - R_\theta(\texttt{prom}, \texttt{compl}_l)\right)$$

is *exactly* maximized. This is precisely the same as maximizing $\sum_{1 \leq i < j \leq n} U\left(r_{\pi_{\texttt{prom}}(i)} - r_{\pi_{\texttt{prom}}(j)}\right)$ over $0 \leq r_1, \ldots, r_n \leq 1$. However, the solution to this optimization program is *independent* of the prompt and, indeed, is the same as the solution to

$$\max_{0 \leq r_1, \ldots, r_n \leq 1} \sum_{1 \leq i < j \leq n} U\left(r_i - r_j\right) \qquad (2)$$

up to a permutation. That is, the empirical distribution of the rewards is independent of the prompt itself in the interpolating regime, thereby leading to reward collapse.

### 2.2 PROMPT-AWARE OPTIMIZATION

To avoid having the same reward distribution, one simple strategy is early stopping. While reward collapse can be avoided via early stopping, early stopping might make the model neglect other important features. A more principled approach is to change the objective. Our proposal is to let the utility function $U$ now depend on the prompt. That is, now we consider training a neural network that maximizes

$$\sum_{(\texttt{prom}, \texttt{compl}_w, \texttt{compl}_l) \in \Pi} U_{\texttt{prom}}\left(R_\theta(\texttt{prom}, \texttt{compl}_w) - R_\theta(\texttt{prom}, \texttt{compl}_l)\right). \qquad (3)$$

In general, the choice of $U_{\texttt{prom}}$ should reflect the open-endedness of the prompt $\texttt{prom}$. An important feature is that if $U_{\texttt{prom}}$ is concave, this problem becomes a convex optimization problem (Lemma 4.1). Given the high flexibility in choosing $U_{\texttt{prom}}$, it is generally recommended to let the practitioners

choose these functions to meet their needs. Nonetheless, below we introduce a family of such functions.

For a strictly increasing utility function $U$, it can be easily demonstrated that the maximum can only be attained when $r_1 \geq \cdots \geq r_n$ (see Lemma B.1 in the Appendix). As a result, we only need to consider the problem

$$\max_{0 \leq r_n \leq \ldots \leq r_1 \leq 1} \sum_{1 \leq i < j \leq n} U\left(r_i - r_j\right). \tag{4}$$

We use the term "reward distribution" to refer to the empirical distribution of solutions to (2) and (4).

**Class 1.** Let $U_\gamma(x) = x^\gamma, x \in [0,1]$ for some $0 < \gamma < 1$. This utility function encourages the reward to take values either near 0 or 1 as $\gamma$ tends to be large. Some plots showing the reward distribution is given in Figure 2(a) and 2(b).

**Class 2.** Let $U_\gamma(x) = -x^\gamma, x \in (0,1]$ for $0 < \gamma \leq 1$ and $U_0(x) = \log x, x \in (0,1]$. We also define $U_\gamma(0) = \infty$ for $0 \leq \gamma \leq 1$. In this case, the reward distribution of Eq. 2 becomes more even as $\gamma$ increases from 0 to 1. Some plots are shown in Figure 2(c) and 2(d).

**Class 3.** Let $U_\sigma(x) = \log \mathtt{sigmoid}(x/\sigma)$ for $\sigma > 0$. The reward distribution becomes more spread between 0 and 1 as $\sigma$ becomes smaller. Some plots are shown in Figure 2(e) and 2(f).

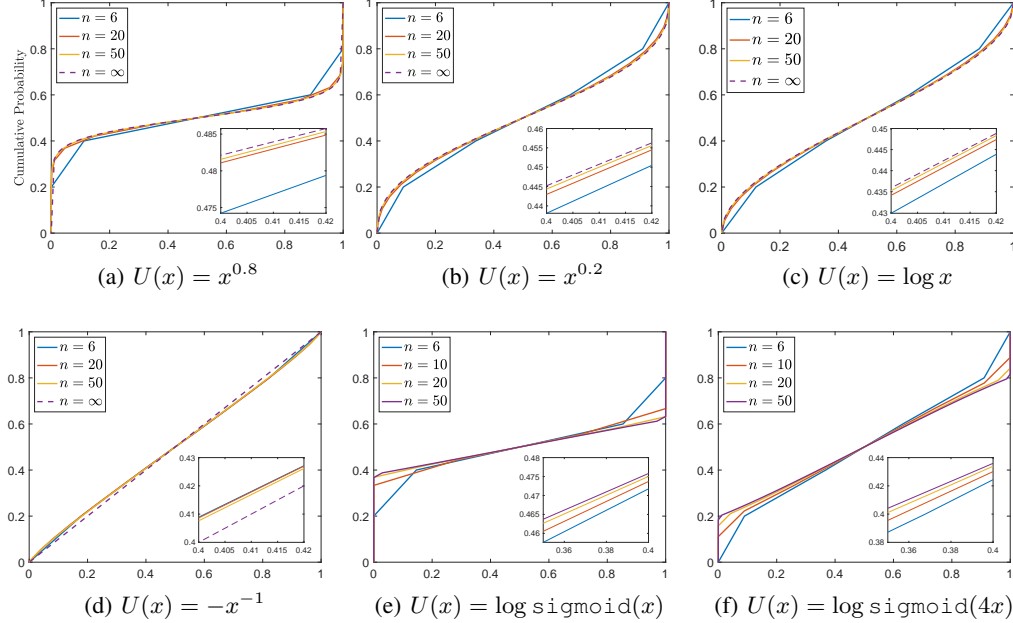

Figure 2: Reward distribution for different utility function.

### 2.3 ASYMPTOTICS

In general, we can explicitly evaluate the reward distribution for any $n$ by solving the optimization (4). Nevertheless, it is helpful to get a handle on the empirical distribution of the solution to this optimization program in the limit $n \to \infty$. The next result gives a closed-form expression of the reward distribution in the case of a large number of completions.

**Theorem 1.** *Let* $U_\gamma(x) = x^\gamma$ *for some* $\gamma \in (0,1)$. *Then the reward distribution of (4) converges to* $\mathrm{Beta}\left(\frac{1-\gamma}{2}, \frac{1-\gamma}{2}\right)$ *as* $n \to \infty$, *which has probability density* $x^{-\frac{1+\gamma}{2}}(1-x)^{-\frac{1+\gamma}{2}}$ *on* $(0,1)$.

**Theorem 2.** *For* $U_\gamma(x) = -x^{-\gamma}$ *for* $0 \leq \gamma \leq 1$ *(as a convention, take* $U_0(x) = \log x$). *Then. the reward distribution of (4) converges in distribution to* $\mathrm{Beta}(\frac{1+\gamma}{2}, \frac{1+\gamma}{2})$.

The proof of Theorem 2 can be found in Martinez-Finkelshtein et al. (2004); Landkof & Landkof (1972). In the limit $\gamma \to 1$ in Theorem 2, the Beta distribution tends to $\mathrm{Beta}(1, 1)$, which is the uniform distribution on $[0, 1]$. This is indeed an example of the one-dimensional Thomson problem (Bowick et al. (2002)), which asks the configuration of $n$ electrons constrained to a line that repel each other with a force given by Coulomb's law. This problem was first considered by Maxwell. Indeed, Martinez-Finkelshtein et al. (2004); Hardin et al. (2004); Amore & Jacobo (2019) prove that the reward distribution will converge to the uniform distribution for $U_\gamma(x) = -x^{-\gamma}$ with $\gamma \geq 1$.

For the above two classes, the limiting distribution does not admit a probability mass. However, probability mass can emerge in the case of a scaled log-sigmoid function.

**Theorem 3.** *If $U$ is strictly increasing and concave, the derivative of the utility function satisfies $U'(0) < \infty, U'(1) > 0$, then the reward distribution of (4) converges in distribution to a probability measure $\mu^*$ that satisfies*

$$\mu^*(\{0\}) = \mu^*(\{1\}) \geq \frac{U'(1)}{U'(0)+U'(1)} > 0.$$

In general, the reward distribution can be characterized from a variational perspective. This gives the following theorem.

**Theorem 4.** *If $U$ is bounded, strongly concave, and increasing. There exists a probability measure $\mu^*$ such that the reward distribution of (2) converges in distribution to $\mu^*$, which is uniquely determined by the following two properties:*

*(a) $\mu^*$ maximizes*

$$\mathbb{E}_{X,X' \overset{iid}{\sim} \mu} U(|X - X'|)$$

*over all probability measures $\mu$ on $[0, 1]$, and*

*(b) it is symmetric with respect to $\frac{1}{2}$ in the sense that, for any measurable set $A \in [0, 1]$ and $1 - A = \{x : 1 - x \in A\}$, $\mu^*(A) = \mu^*(1 - A)$.*

## 3 EXPERIMENTS

In this section, we conduct experiments to investigate the phenomenon of reward collapse and demonstrate that prompt-aware training can prevent reward collapse.

### 3.1 EVIDENCE OF REWARD COLLAPSE IN LARGE LANGUAGE MODEL

We start our investigation by conducting experiments utilizing a LLM, specifically GPT-Neo-1.3B (Black et al. (2021)). Guided by the methodologies outlined in the StackLlama project (Beeching et al. (2023)), we trained the model on the StackExchange preference dataset (Lambert et al. (2023)), a robust resource that provides rankings of responses for individual prompts.

Constrained by computational resources, we focused our training on a carefully selected subset of the dataset containing only the prompts accompanied by exactly five responses. Our experimental setup comprised 128 distinct prompts, each of which contributed 10 pairs to the reward modeling process. By adopting the codebase from StackLlama (Beeching et al. (2023)), and setting the learning rate to $3 \times 10^{-5}$ along with a batch size of 20 pairs, we carried out the training over 10 epochs.

As demonstrated in Figure 1, our results highlight the emergence of the reward collapse phenomenon under these realistic conditions. The evidence of this effect can be observed as the distribution becomes increasingly concentrated over the course of the training.

### 3.2 SETUP OF OUR SECOND EXPERIMENT

The open-source datasets currently available for RLHF are rather limited. Most of these datasets (Nakano et al. (2021); Bai et al. (2022a)) typically include only a handful of candidate responses (usually a single pair) for each corresponding prompt question. Moreover, the ranking signals in those datasets are usually noisy, either because they are sourced from the Internet (Ethayarajh et al. (2023)) or because of the inherent subjectivity of the ranking process.

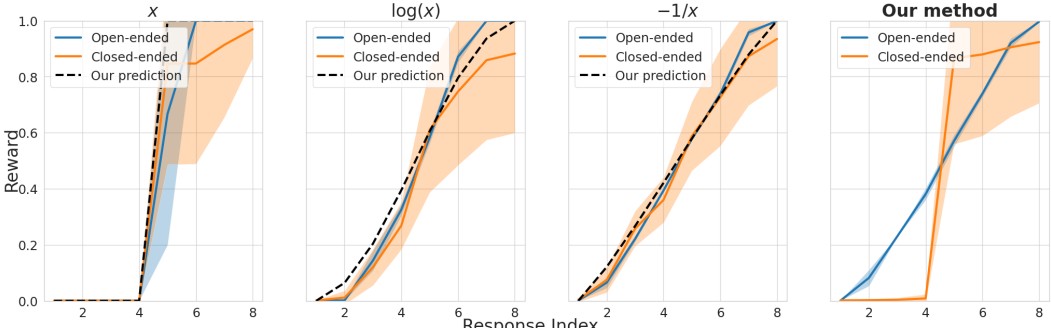

Figure 3: **Reward collapse on the test set.** The reward distributions have similar collapse phenomenons on the test set, and using prompt-aware loss can mitigate the collapse.

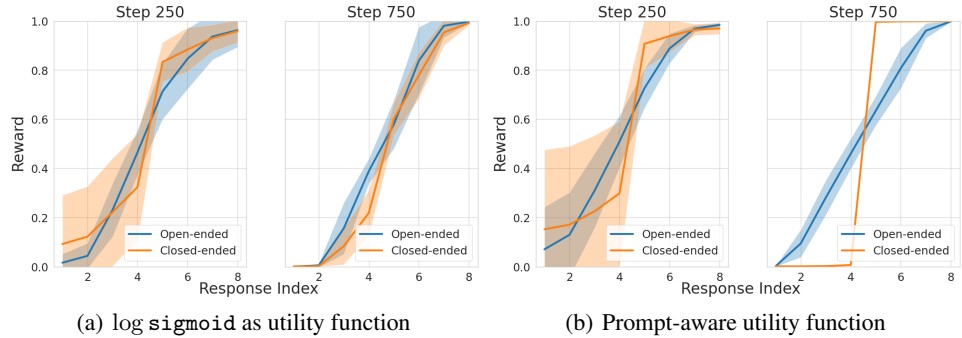

(a) $\log$ `sigmoid` as utility function        (b) Prompt-aware utility function

Figure 4: **(Left)** The reward distribution of different prompts gradually converges into a single distribution during training. **(Right)** When using the prompt-aware loss function, the reward distributions of the two different prompts can be gradually separated during training.

In order to conduct a carefully controlled experiment, we curated our own dataset, focusing on a single, simplified feature – the length of the response, measured in terms of word count as the ground truth reward. A subset of questions was selected from the LongForm dataset (Köksal et al. (2023)), a question-answer dataset characterized by its lengthy answers. To simulate scenarios with open-ended and concrete problems, we truncated the original answer according to two distinct length distributions, thereby generating eight responses for each prompt: the first distribution is nearly uniform, ranging from 10 to 80 words, while the second is a polarized distribution with response lengths primarily clustered around either 30 or 60 words. Each question was randomly assigned as either open-ended or concrete. [3] Additionally, the phrases "Write the answer in an open-ended way." and "Write either a short answer or a long answer." were added to the open-ended and concrete questions, respectively, to distinguish the question type. Following this process, we constructed a dataset comprising 8192 training questions and 16 test questions.

In our experiments, we focus on the following $U$ functions: $x$, $\log x$, $-1/x$, as well as $\log$ `sigmoid`$(x)$, which is employed in Ouyang et al. (2022) and the prompt-aware $U$, which adaptively selects $U$ from $x$ and $-1/x$. Given that the $U$ function operates on $x$ in the range $[-1, 1]$, we adjust some $U$ functions with suitable continuous extensions or scaling. We then train a DeBERTa V3 (He et al. (2021)) as the reward model. The training details can be found in Appendix A.1.

### 3.3 EXPERIMENTAL RESULTS

**Fixed loss function leads to reward collapse.** As depicted in Figure 4(a), reward distributions corresponding to different prompts gradually converge towards a single, prompt-independent distribution throughout the training process. Specifically, in the context of Figure 4(a), where the $U$ function is represented by `LogSigmoid`, the reward distribution exhibits positive probability mass at reward

---

[3]In practice, such assignments can be done by various methods. See Appendix A.2 for a short discussion.

scores of 0 and 1 (illustrated by the flat segments corresponding to the first two and last two scores). This observation validates the prediction encapsulated in Theorem 3. Examining other $U$ functions, Figures 3 collectively indicates the occurrence of loss collapse on the test datasets. Specifically, employing $x$ as the $U$ function results in a polarized reward distribution, whereas utilizing $-1/x$ as the $U$ function yields a uniform reward distribution.

**Prompt-aware training avoids reward collapse.** Figures 3 shows the reward distribution at the end of training with varying utility functions. The results along with Figure 4(b) reveal that using a prompt-aware $U$ function effectively prevents reward collapse across both training and test datasets. This strategy yields a more uniform reward distribution for open-ended prompts while promoting a more polarized reward distribution for concrete prompts.

## 4 PROOFS

In this section, we will briefly present the proofs of results in Section 2. However, we will deviate from the previous order and start by proving Theorem 4. We also put the proof of Theorem 3 into Appendix C.3 due to the length constraint. Let

$$S(r_1, \cdots, r_n) := \sum_{1 \leq i < j \leq n} U(r_i - r_j) \text{ and } \hat{\mathbf{r}} \equiv (\hat{r}_1, \ldots, \hat{r}_n) := \arg \max_{0 \leq r_1, \cdots, r_n \leq 1} S(r_1, \cdots, r_n).$$

In addition, for any vector $(u_1, \cdots, u_n) \in \mathbb{R}^n$, we employ boldface notation $\mathbf{u}$ to represent the entire vector. THis allows us to write $S(\mathbf{r})$.

### 4.1 PROOF OF THEOREM 4

First, when $U$ is concave and strictly increasing, $\hat{\mathbf{r}}$ exhibits the following properties:

**Lemma 4.1.** *If $U$ is strictly concave and strictly increasing, the function $S(\mathbf{r})$ is concave. Therefore, the optimization problem uniquely determines $\hat{\mathbf{r}}_n$. Additionally, the following properties hold: (1) $\hat{r}_1 \geq \cdots \geq \hat{r}_n$, and (2) $1 - \hat{r}_i = \hat{r}_{n-i+1}$ for any $1 \leq i \leq n$.*

The proof of Lemma 4.1 is straightforward and is provided in Appendix B.1. Upon further examination of the function $S(\mathbf{r})$, we discover that if $U$ is strongly concave with parameter $\mu > 0$, then $S$ also exhibits some kind of strongly concavity, except in the direction $(1, 1, \cdots, 1)$. This property is formulated in the following lemma.

**Lemma 4.2.** *If $U$ is strongly concave with parameter $\mu > 0$, and we consider another vector $\mathbf{u} = (u_1, \ldots, u_n)$ where $u_1 \geq \cdots \geq u_n$, the following inequality holds:*

$$S(\mathbf{u}) - S(\hat{\mathbf{r}}) \leq -\frac{n\mu}{2} \| \operatorname{Proj}_{V_n}(\mathbf{u} - \hat{\mathbf{r}}) \|^2.$$

*Here, $V_n \subset \mathbb{R}^n$ is the subspace orthogonal to $(1, \cdots, 1)$, and $\| \cdot \|$ represents the Euclidean norm.*

The proof of this lemma can be found in Appendix B.2. Our next lemma quantifies the difference between two symmetric probability measures.

**Lemma 4.3.** *For two different symmetric probability measure $\mu_1$ and $\mu_2$ on $[0, 1]$, let $r_i^{(j)} = \frac{1}{2} \inf\{t : \mu_j([0, t]) \geq \frac{n-i}{n-1}\} + \frac{1}{2} \sup\{t : \mu_j([0, t)) < \frac{n-i}{n-1}\}), i = 1, 2, \cdots, n; j = 1, 2$. Then there exists positive constant $c_0$ such that for all $n$,*

$$\| \operatorname{Proj}_{V_n}(\mathbf{r}^{(1)} - \mathbf{r}^{(2)}) \|_2^2 \geq c_0 n.$$

The proof of Lemma 4.3 is also provided in Appendix B.3. Now, we are ready to prove the uniqueness part of Theorem 4. Due to the length constraint, we will present it as a separate lemma and defer the proof to Appendix B.4. In short, we use Lemma 4.2 and 4.3 to demonstrate that for two distinct symmetric measures, their distance is sufficiently large such that at least one of them is not optimal.

**Lemma 4.4.** *If $\mu_1$ and $\mu_2$ are two symmetric probability measure which both maximize*

$$\mathbb{E}_{X, X' \overset{iid}{\sim} \mu} U(|X - X'|)$$

*over all probability measures $\mu$ on $[0, 1]$. Then we have $\mu_1 = \mu_2$.*

Now we are ready to prove the convergence part of Theorem 4.

*Proof of Theorem 4.* Let $\hat{\mathbb{P}}_n := \frac{1}{n}\sum_{i=1}^n \delta_{\hat{r}_n}$ denote the empirical distribution of $\hat{\mathbf{r}}_n$. Note that $\{\hat{\mathbb{P}}_n\}$ are probability measures defined on $[0,1]$, so they are tight. By Prohorov's theorem, there exists a sub-sequence $\{k(n)\}_{n\geq 1}$ such that $\hat{\mathbb{P}}_{k(n)} \xrightarrow{d} \hat{\mu}$. Let $X_n, X'_n \overset{iid}{\sim} \hat{\mathbb{P}}_n$ and $\hat{X}, \hat{X}' \overset{iid}{\sim} \hat{\mu}$. By continuous mapping theorem, we also have $|X_n - X'_n| \xrightarrow{d} |\hat{X} - \hat{X}'|$. Moreover, because $U$ is bounded and continuous, Portmanteau theorem gives

$$\mathbb{E}_{X,X'\overset{iid}{\sim}\hat{\mathbb{P}}_{k(n)}} U(|X-X'|) \to \mathbb{E}_{X,X'\overset{iid}{\sim}\hat{\mu}} U(|X-X'|).$$

Let $\mu$ be another probability measure on $[0,1]$. Let $\hat{\mathbb{Q}}_n = \frac{1}{n}\sum_{i=1}^n \delta_{q_{n,i}}$ such that $\hat{\mathbb{Q}}_n \xrightarrow{d} \mu$. By the same argument before, we also have $\mathbb{E}_{X,X'\overset{iid}{\sim}\hat{\mathbb{Q}}_{k(n)}} U(|X-X'|) \to \mathbb{E}_{X,X'\overset{iid}{\sim}\mu} U(|X-X'|)$. Then by the optimal assumption of $\hat{\mathbf{r}}_n$,

$$\begin{aligned}
\mathbb{E}_{X,X'\overset{iid}{\sim}\hat{\mu}} U(|X-X'|) &= \lim_{n\to\infty} \mathbb{E}_{X,X'\overset{iid}{\sim}\hat{\mathbb{P}}_{k(n)}} U(|X-X'|) \\
&\geq \lim_{n\to\infty} \mathbb{E}_{X,X'\overset{iid}{\sim}\hat{\mathbb{Q}}_{k(n)}} U(|X-X'|) = \mathbb{E}_{X,X'\overset{iid}{\sim}\mu} U(|X-X'|).
\end{aligned}$$

This means $\hat{\mu}$ maximize $\mathbb{E}_{X,X'\overset{iid}{\sim}\mu} U(|X-X'|)$ over all probability measure $\mu$ on $[0,1]$. From Lemma 4.1, we know that $1 - \hat{r}_i = \hat{r}_{n-i+1}$, so $\hat{\mu}$ is symmetric. If there is another sub-sequence $m(n)$ such that $\hat{\mathbb{P}}_{m(n)} \xrightarrow{d} \hat{\nu}$. By the same argument before, $\hat{\nu}$ is also optimal and symmetric. From Lemma 4.4, $\hat{\mu} = \hat{\nu}$. Thus for every converging sub-sequence of $\{\hat{\mathbb{P}}_n\}$, the limit distribution must be the same. By the tightness of $\{\hat{\mathbb{P}}_n\}$, we have $\hat{\mathbb{P}}_n \xrightarrow{d} \mu^*$. □

### 4.2 PROOF OF THEOREM 1

For the utility function $U_\gamma(x) = x^\gamma$, having established Theorem 4, our objective is to identify a symmetric probability measure $\mu^*$ that maximizes $\mathbb{E}_{X,X'\overset{iid}{\sim}\mu} U_\gamma(|X-X'|)$. By employing the variational principle, we can derive a condition that is necessary for optimality. Notably, this condition also suffices for optimality.

**Lemma 4.5.** *Let $U_\gamma(x) = x^\gamma$ for some $\gamma \in (0,1)$. A probability measure $\mu$ on $[0,1]$ will maximize $\mathbb{E}_{X,X'\overset{iid}{\sim}\mu} U_\gamma(|X-X'|)$ if it satisfies the condition that $\mathbb{E}_{X\sim\mu} U_\gamma(|X-c|)$ is independent of $c \in [0,1]$.*

The proof of Lemma 4.5 is provided in Appendix C.1. Therefore, proving Theorem 1 is reduced to verifying the condition stated in Lemma 4.5. This verification process is tedious and will be deferred to Appendix C.2 for brevity.

## 5 EXTENSION TO PAIRWISE COMPARISONS

Our Prompt-Aware approach can be generalized to accommodate other settings, such as instances where only pairwise preference data is accessible. Pairwise preference data may include loops, similar to the rock-paper-scissors scenario, and can be produced from a probabilistic model. Consequently, the data might simultaneously indicate a preference of A over B and a preference of B over A. Pairwise preference data is extensively utilized in RLHF (Christiano et al. (2017); Ibarz et al. (2018); Ziegler et al. (2019); Ouyang et al. (2022); Zhu et al. (2023)).

We explore the well-known Bradley-Terry-Luce (BTL) model (Bradley & Terry (1952); Luce (2012)), which assumes the existence of scores $\{\theta_i\}_{1\leq i\leq n}$ for $n$ items such that the preference between item $i$ and item $j$ is given by $\mathbb{P}(i \text{ is preferred over } j) = \sigma(\theta_i - \theta_j)$, where $\sigma$ denotes the sigmoid function $\sigma(x) = 1/(1 + \exp(-x))$. This probabilistic model effectively captures the relative preferences between items, based on the disparity in their underlying scores.

To illustrate our framework, we consider the following expected version problem:

$$\max_{0\leq r_1,\cdots,r_n\leq 1} S(r_1,\cdots,r_n), \text{ where } S(r_1,\cdots,r_n) = \sum_{1\leq i,j\leq n} U(r_i - r_j)\sigma(\theta_i - \theta_j).$$

The function $S(\mathbf{r})$ is similar to a family of log-likelihood functions considered in (Noothigattu et al. (2020)). We presume that $U$ is increasing and concave. Then similar to Lemma 4.1, $U$ is also concave in $(r_1, \cdots, r_n)$. Let $\hat{\mathbf{r}} = (\hat{r}_1, \ldots, \hat{r}_n)$ be the vector that maximizes $S(\mathbf{r}) = \sum_{1 \leq i,j \leq n} U(r_i - r_j)\sigma(\theta_i - \theta_j)$. We present the following consistency result on $\hat{\mathbf{r}}$:

**Theorem 5.** *Assuming that $U$ is increasing and strongly concave with a constant $\mu > 0$ and $\kappa = \max_{1 \leq i \leq n} |\theta_i|$. Then $\hat{\mathbf{r}}$ keep the order of $\{\theta_i\}_{1 \leq i \leq n}$, and we have the following:*

$$|\hat{r}_i - \hat{r}_j| \leq 2\sqrt{U(1)(1 + e^\kappa)|\theta_i - \theta_j|/\mu}.$$

The proof of these results can be found in Appendix D. Theorem 5 ensures that for any increasing and strongly concave utility function $U$, $\hat{\mathbf{r}}$ is a reliable estimate of $\{\theta_i\}_{1 \leq i \leq n}$, in the sense that $\hat{r}_i$ and $\hat{r}_j$ are close if $\theta_i$ and $\theta_j$ are close.

Even though we may not be able to determine the precise limiting distribution of $\mathbf{r}_n$ in this extended setting, we can still extract insights from our previous analysis in Section 2. As previously observed, selecting $U(x) = x$ tends to polarize the reward distribution, while selecting $U(x) = -1/x$ yields a more uniform reward distribution. This phenomenon is also evident in this setting, as observed in the results presented in Figure 5. More details is given in Appendix D.

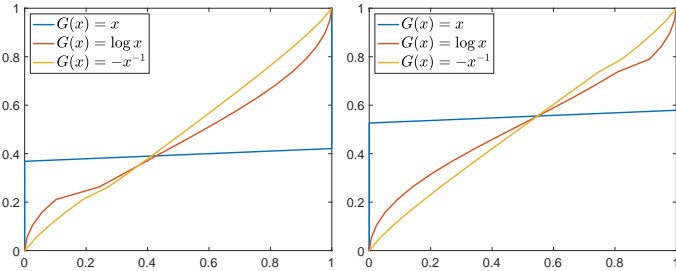

Figure 5: Reward distribution with different choice of $\{\theta\}_{1 \leq i \leq n}$ when $n = 21$.

Based on these findings, we can conclude that in this extended setting, we can also employ a prompt-aware utility function $U$ to mitigate reward collapse and achieve the desired reward distribution by carefully selecting the form of $U$. This provides us with flexibility in shaping the reward distribution according to our specific requirements.

## 6 DISCUSSION

In this paper, we have introduced an empirical phenomenon known as reward collapse that arises during reward model training for aligning LLMs using human preference rankings. This phenomenon results in the same reward distribution regardless of the prompt type. The occurrence of reward collapse stems from neural network interpolation during the final training phase. To mitigate reward collapse, we propose utility functions that consider the nature of prompts and an analytical framework that evaluates reward distribution, yielding closed-form reward expressions. Synthetic experiments substantiate our findings, presenting a method superior to early stopping to tackle reward collapse.

While our experiments provide valuable insights, it is important to acknowledge their limitations, primarily stemming from the constrained computational resources available. Given abundant resources, future research can explore the use of a more diverse range of prompts, varying in terms of their open-endedness. Additionally, it would be interesting to investigate the extent to which the trained reward model enhances the capabilities of large language models, such as their ability to self-calibrate uncertaintycite (Lin et al. (2022); Kadavath et al. (2022)). Theoretical investigations could focus on finding increasing, concave functions that precisely match a given discrete reward distribution. On the practical side, developing a method to choose a utility function based on prompts, perhaps using a parameter such as $\gamma$ in Section 2.2, poses an intriguing avenue for further exploration. Furthermore, exploring the potential benefits of truncated ranking by requiring human labelers to provide partial rankings of acceptable completions and ignore unacceptable completions could offer valuable insights into improving the training of reward models.

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
