# OpenReview forum: "Reward Collapse in Aligning Large Language Models"
_ICLR.cc/2024/Conference — Submitted to ICLR 2024_

### Official Review · Reviewer_vqeA · 2023-10-18

**Soundness:** 2 fair
**Presentation:** 2 fair
**Contribution:** 2 fair
**Rating:** 5
**Confidence:** 3

**Summary:**

The paper study the problem of RLHF for large language models. The paper firstly finds that the reward model trained by human rankings over different completions can not distinguish with open-ended and closed prompts, and call it as  the "reward collapse" phenomenon. The paper theoretically gives the reason, that is, the ranking-based objective function does not consider prompt-based factors. Some experimental analysis validates the claim.

**Strengths:**

Novelty: The claim proposed by the paper is novel and interesting.
Quality: The paper gives enough theoretical analysis and the experimental results validates the phenomenon.
Clarity: The paper is well written.
Significance: The proposed claim is interesting and meaningful for the LLM community.

**Weaknesses:**

Quality: The paper claims that we should use prompt-based utility function Eq.(3). However, all follow analysis is about the utility function without the prompt, which is inconsistent. The paper does not provide the implementation details of the prompt-aware training.

**Questions:**

See the above section.

---

> ### Author Response · Authors · 2023-11-20
> **Response to Reviewer vqeA**
>
> Thank you for your valuable feedback! We respond to your comments as follows:
>
> > The paper claims that we should use prompt-based utility function Eq.(3). However, all follow analysis is about the utility function without the prompt, which is inconsistent.
>
> Our theoretical analysis focuses on determining the reward distribution when a fixed utility function is chosen. Therefore, for the purposes of our analysis, we can fix a utility function.
>
> > The paper does not provide the implementation details of the prompt-aware training.
>
> In our experiment in Section 3.2, we randomly assigned each question as either open-ended or concrete. For open-ended questions, the utility function is selected as G(x) = -1/x. For concrete questions, their utility function is chosen as G(x) = x. This choice of utility functions serves the specific purpose of demonstrating that the prompt-aware approach can effectively prevent reward collapse. Additionally, in Appendix A.2, we delve into potential practical methods for assigning prompt types.

---

### Official Review · Reviewer_hn6B · 2023-10-31

**Soundness:** 4 excellent
**Presentation:** 4 excellent
**Contribution:** 3 good
**Rating:** 6
**Confidence:** 4

**Summary:**

This paper studies the reward modeling in the alignment for LLMs. Specifically, the authors investigate the phenomenon of reward collapse in LLMs and proposes a prompt-aware optimization scheme to mitigate it. The key idea of the paper is to use a prompt-dependent utility function so that the distributions can be more diverse across different prompts. Empirical results are provided to verify the theoretical and intuitive results.

**Strengths:**

1 This paper is about an important and interesting problem in RLHF, and is very relevant to the community of Neurips. The authors first demonstrate the problem with various experiments of reward modeling, and motivate the approach with sound theoretical analysis.

2 The paper proposes a prompt-aware optimization scheme to overcome reward collapse and introduces utility functions that depend on the prompt to achieve prompt-dependent reward distributions. Real-world experiments on stackoverfolow and also synthetic experiments are conducted to support the findings and demonstrate a method superior to early stopping for addressing reward collapse.

As a paper for understanding some important problem in RLHF, the quality of the work is satisfactory.

**Weaknesses:**

1 How do the conclusions change if the assumption that the LLM is sufficiently overparameterized so that it can maximize the utility for all the prompts (discussions around equation (2))?

2 While the story and theoretical analysis are sound, the evidences of this paper are limited. But one thing I believe can largely improve the paper is that we can further evaluate the quality of the reward model by best-of-n policy. Specifically, we can fix a LLM, and for each prompt, we sample n responses and then take the one with the highest reward as the final output. Then, we can compare the responses by either human evaluation or GPT4 evaluation. For more details, you may check [1].

------------------------------------
Update in 11.11
Sorry for the late update. I just read the paper again and have a quick question about the choice of U. As I mentioned in the above review, the experiments conducted are rather limited and simple. In particular, the utility function used for the response length seems to be hard to generalize to general practical applications. Could you give an example of the choice of utility function in practice, e.g., used for the HH-RLHF dataset (whose details can be found in huggingface).

[1] let's verify step by step

**Questions:**

see weakness

---

> ### Author Response · Authors · 2023-11-20
> **Response to Reviewer hn6B**
>
> Thank you for your support of our paper. We address the questions as follows.
>
> 1.	The assumption that the LLM is sufficiently overparameterized is quite strong, but it’s essential for our conclusions. Since if the reward does not maximize the objective function exactly, reward collapse may not exists. For example, in the experiment in Section 3 (Figure 1 and 4), reward collapse does not show up in the early stage of training. This overparameterization assumption is satisfied in most real applications. For example, InstructGPT  se a 6B reward model [1].
>
> 2.	While we acknowledge the potential improvement of RLHF in current LLMs, the primary focus of this paper is to document and investigate the phenomenon of reward collapse during reward model training in LLMs. We present theoretical results on reward collapse and derive the reward distribution under a specific class of utility functions. Although intuitively our approach should enhance the performance of RLHF, we believe that the impact of using a prompt-aware utility function requires thorough further research. Numerous techniques implicitly avoid the harm of reward collapse, such as early-stop or adding regularizers, which may alleviate reward collapse. Investigating the extent to which the trained reward model enhances the capabilities of large language models, such as their ability to self-calibrate uncertainty, is an interesting direction for future work beyond the scope of this paper.
>
> 3.	Aassigning prompt types is fundamental in the prompt-aware approach, as discussed in Appendix A.2. A straightforward method for determining the prompt type is to gather input from human labelers, who typically rank different responses, as seen in InstructGPT. Additionally, we can request them to assess how open-ended the prompt is using a scale ranging from -1 to 1. Automated annotation processes are also possible. For instance, one approach involves assessing the variability of responses to a given prompt. If the responses exhibit high variability, the prompt can be classified as open-ended. Conversely, if the responses show low variability, the prompt may be deemed close-ended. Determining the prompt type is indeed a complex and fascinating task, providing a promising avenue for future research.
>
> [1] Ouyang, Long, Jeffrey Wu, Xu Jiang, Diogo Almeida, Carroll Wainwright, Pamela Mishkin, Chong Zhang et al. "Training language models to follow instructions with human feedback." Advances in Neural Information Processing Systems 35 (2022): 27730-27744.
>
> We sincerely value your feedback, and we hope these clarifications address your concerns. If you have any further questions or suggestions, please feel free to let us know.

---

### Official Review · Reviewer_C1CM · 2023-11-01

**Soundness:** 2 fair
**Presentation:** 2 fair
**Contribution:** 2 fair
**Rating:** 5
**Confidence:** 3

**Summary:**

This paper presents the theoretical finding of reward collapse when training reward models on ranking-based preference data. Through experiments, the authors demonstrate that the reward distributions for different prompts converge to a common prompt-independent distribution, disregarding whether prompts are open or closed-ended. To address this issue, they propose a prompt-aware utility function approach that learns distinct reward distributions based on prompt type.

**Strengths:**

strengths:
1. The paper clearly documents the phenomenon of reward collapse, supported by theoretical analysis and experiments. This is an important observation that will aid the development of prompt-aware reward modeling.
2. Empirically demonstrated that the reward distributions will converge towards a prompt-independent distribution.
3. The method is extended to handle pairwise preference data, improving applicability.

**Weaknesses:**

1. Does not provide much detail on how the prompt-aware utility function U_prom adaptively selects between U(x) = x and U(x) = -1/x in the experiments mentioned in Section 3.2. Do you manually assign utility functions based on the question type?
2. Experiments are done on only synthetic datasets where the word count is the ground-truth reward. Real-world ranking datasets would provide stronger validation.
3. It is not clear how the prompt-dependent reward distribution will contribute to the performance increase for RLHF or other direct optimization methods or just the best of n sampling. It would be great to see how this reward distribution will increase performance.
4. The evidence of the reward collapse in Figure 1: what does the dashed region represent? is it over the 128 prompt datasets? It would be clearer to present the reward collapse phenomenon on a close-end question.
5. For the two subfigures in Figure 5, what are the differences between the two plots’s settings? Could you label the y-axis?

**Questions:**

please see weaknesses

---

> ### Author Response · Authors · 2023-11-20
> **Response to Reviewer C1CM**
>
> Thank you for taking the time to provide feedback on our work. We appreciate your thoughtful comments and would like to address each point accordingly.
> 1.	 In Section 3.2, we specified that each question was randomly assigned as either open-ended or concrete. For open-ended questions, their utility function is chosen as G(x) = -1/x. For concrete questions, their utility function is chosen as G(x) = x. This is only for the purpose of our experiment. Additionally, in Appendix A.2, we delve into potential practical methods for assigning prompt types.
>
> 2.	We understand your concern about the validation of our findings on real-world datasets. While Section 3.1 presented experiments on a real-world dataset to demonstrate the existence of reward collapse, we used synthetic datasets in Section 3.2 to illustrate how prompt-aware training can mitigate this issue. We acknowledge the potential for stronger validation with real-world ranking datasets and agree that it is an area for improvement. Nevertheless, our controlled experiments, combined with those in Section 3.1, effectively convey the key message: reward collapse can be avoided through prompt-aware training.
>
> 3.	The primary focus of our paper is to document and investigate the phenomenon of reward collapse in the training of reward models for LLMs. We have developed theoretical results on reward collapse and derived the reward distribution under a specific class of utility functions. While we intuitively believe that our approach enhances the performance of RLHF, we recognize the need for further research on the effects of using prompt-aware utility functions. We also acknowledge the existence of other techniques that implicitly address reward collapse, such as early-stop or regularizers. Exploring how the trained reward model enhances the capabilities of large language models, including their ability to self-calibrate uncertainty, is an intriguing avenue for future research, albeit beyond the scope of this paper.
>
> 4.	Yes, the dashed region signifies the randomness across 128 distinct prompts. We appreciate your suggestion, and in response, we will employ an alternative utility function to ensure a clearer presentation of the results.
>
> 5.	The distinction lies in the utilization of two different parameters, denoted as $\theta$ (Appendix D.2), in the BTL model. The y-axis represents the value of the empirical cumulative distribution function (ecdf) of rewards obtained from solving the optimization problem:
> $$
> \max_{0\le r_1,\cdots, r_n \le 1} S(r_1,\cdots,r_n), \mbox{ where }S(r_1,\cdots,r_n) = \sum_{1\le i,j\le n} U(r_i - r_j) \sigma(\theta_i - \theta_j).
> $$
> We will ensure that these details are appropriately highlighted for better clarity in our presentation.
>
> We sincerely value your feedback, and we hope these clarifications address your concerns. If you have any further questions or suggestions, please feel free to let us know.

---

### Meta-Review · Area_Chair_64K4 · 2023-12-05

**Metareview:**

The paper identifies 'reward collapse' in LLMs, where reward models trained on ranking-based preference data converge to a uniform reward distribution for different prompts. It proposes a prompt-aware optimization scheme using diverse utility functions to address this, supported by theoretical and empirical analyses. Strengths include identification of the reward collapse phenomenon, theoretical analysis, and an initial attempt of using prompt-dependent utility functions for diverse reward distributions.

Weaknesses: Limited validation on commonly-used / real-world preference datasets, as well as lack of clarity on the impact of reward collapse on downstream RLHF performance. Additionally, reviewers raised concerns about the practical applicability of prompt-aware utility functions. Since reward collapse is an empirical phenomenon, experiments primarily focused on synthetic datasets is convincing enough that this phenomenon matters in practice.

**Justification For Why Not Higher Score:**

The paper straddles the acceptance threshold, with its contributions recognized as relevant and significant but requiring additional clarity and empirical validation on common preference datasets in RLHF.

**Justification For Why Not Lower Score:**

N/A

---

### Decision · Program_Chairs · 2024-01-16

Reject